# Tailoring the Properties of Thermo-Compressed Polylactide Films for Food Packaging Applications by Individual and Combined Additions of Lactic Acid Oligomer and Halloysite Nanotubes

**DOI:** 10.3390/molecules25081976

**Published:** 2020-04-23

**Authors:** Sandra Rojas-Lema, Luis Quiles-Carrillo, Daniel Garcia-Garcia, Beatriz Melendez-Rodriguez, Rafael Balart, Sergio Torres-Giner

**Affiliations:** 1Technological Institute of Materials (ITM), Universitat Politècnica de València (UPV), Plaza Ferrándiz y Carbonell 1, 03801 Alcoy, Spain; sanrole@epsa.upv.es (S.R.-L.); luiquic1@epsa.upv.es (L.Q.-C.); dagarga4@alumni.upv.es (D.G.-G.); rbalart@mcm.upv.es (R.B.); 2Novel Materials and Nanotechnology Group, Institute of Agrochemistry and Food Technology (IATA), Spanish National Research Council (CSIC), Calle Catedrático Agustín Escardino Benlloch 7, 46980 Paterna, Spain; beatriz.melendez@iata.csic.es

**Keywords:** PLA, OLA, HNTs, ultrasound-assisted dispersion, nanocomposites, food packaging

## Abstract

In this work, films of polylactide (PLA) prepared by extrusion and thermo-compression were plasticized with oligomer of lactic acid (OLA) at contents of 5, 10, and 20 wt%. The PLA sample containing 20 wt% of OLA was also reinforced with 3, 6, and 9 parts per hundred resin (*phr*) of halloysite nanotubes (HNTs) to increase the mechanical strength and thermal stability of the films. Prior to melt mixing, ultrasound-assisted dispersion of the nanoclays in OLA was carried out at 100 °C to promote the HNTs dispersion in PLA and the resultant films were characterized with the aim to ascertain their potential in food packaging. It was observed that either the individual addition of OLA or combined with 3 *phr* of HNTs did not significantly affect the optical properties of the PLA films, whereas higher nanoclay contents reduced lightness and induced certain green and blue tonalities. The addition of 20 wt% of OLA increased ductility of the PLA film by nearly 75% and also decreased the glass transition temperature (*T_g_*) by over 18 °C. The incorporation of 3 *phr* of HNTs into the OLA-containing PLA films delayed thermal degradation by 7 °C and additionally reduced the permeabilities to water and limonene vapors by approximately 8% and 47%, respectively. Interestingly, the highest barrier performance was attained for the unfilled PLA film plasticized with 10 wt% of OLA, which was attributed to a crystallinity increase and an effect of “antiplasticization”. However, loadings of 6 and 9 *phr* of HNTs resulted in the formation of small aggregates that impaired the performance of the blend films. The here-attained results demonstrates that the properties of ternary systems of PLA/OLA/HNTs can be tuned when the plasticizer and nanofiller contents are carefully chosen and the resultant nanocomposite films can be proposed as a bio-sourced alternative for compostable packaging applications.

Academic Editors: Alfonso Jimenez and María del Carmen Garrigós

## 1. Introduction

Most commercial plastic packaging is currently made of non-biodegradable polymers derived from petroleum that is currently causing disposal problems and leakage into the oceans [1]. To solve these environmental issues, recent studies have been directed towards the development osf food packaging materials based on biopolymers that have the two-fold environmental advantage of being bio-based, that is, obtained from renewable resources, and biodegradable, that is, disintegrable in controlled compost soil due to their potentially hydrolysable ester bonds [2]. Among them, polylactide (PLA) is nowadays considered the front-runner in the emerging bioplastics market, showing an annual consumption of 140,000 tons within a bioplastics market that is growing by 20–30% each year [3]. Moreover, PLA as well as other thermoplastic biopolymers such polyhydroxyalkanoates (PHAs) and thermoplastic starch (TPS) present good thermal and mechanical properties and they can be melt processed by film and sheet extrusion, injection molding, thermoforming, foaming, fiber spinning, etc. [4,5]. In particular, the physical properties of PLA are fairly similar to those of petrochemical-based polymers with high strength and low toughness, such as polystyrene (PS) and polyethylene terephthalate (PET) [6]. Current applications of PLA in the food packaging industry include food trays, disposable glasses for cold drinks, lids, and straws, among others [7].

However, in general terms, biopolymers can be either strongly plasticized by sorption of moisture or show, for instance, low-to-medium barrier properties to gases and aroma [8]. In particular, the main performance drawbacks of PLA are mainly associated to its low thermal resistance, excessive brittleness, and insufficient barrier to oxygen and water compared to other benchmark packaging polymers like PET [9]. It is, thus, of great industrial interest to increase the overall performance of biopolymers in order to generate ‘added value’ arguments to counteract their current pricing and enable the substitution for more established petrochemical polymers. In this context, the use of novel plasticizers and nanoclays has a significant potential to enhance the physical performance of PLA materials. In this context, lactic acid oligomer (OLA) has been proposed as an alternative to common plasticizers for producing flexible films of PLA by taking advantage of their similar chemical structure and renewable origin [10]. Thus, amounts from 10 to 30 wt% of OLA are required to provide a substantial reduction of the PLA’s glass transition temperature (*T_g_*) as well as to obtain adequate mechanical properties for films manufacturing [11,12]. For instance, it was reported that 20 wt% of OLA improves substantially ductility with a significant decrease in *T_g_* [13].

In the large field of nanotechnology, polymer matrix based nanocomposites have become a prominent area of current research and development [14]. There are several types of nanoclays that can be utilized to prepare PLA nanocomposites such as montmorillonite (MMT), sepiolite, cellulose nanowhiskers (CNWs), graphene nanoplatelets (GNPs), carbon nanotubes (CNTs) or halloysite nanotubes (HNTs), among others. [15]. Nanofillers can promote higher improvements at lower contents (≤5 wt%) over classical fillers due to their high aspect ratio since at least one filler dimension is below 100 nm. This morphology renders a high surface area by which their reinforcement efficiency can match that of microcomposites with 40–50 wt% of loadings. Although HNTs have been less studied than layered silicates, this nanoclay offers unique properties for food packaging applications such as natural origin, abundant availability, low density, non-toxic characteristics, low cost (~0.45 USD/g), and high rigidity [16]. More interestingly, HNT offers easy dispersability due to the low density of hydroxyl (–OH) groups on the surface [17]. This nanoclay consists on natural aluminosilicate (Al_2_Si_2_O_5_(OH)_4_·2H_2_O) crystallites in which the hollow tubular morphology is formed by layer rolling. In particular, HNT displays a high aspect ratio with nanotube length up to 15 μm with internal and external diameters in the ranges of 10–100 nm and 20–200 nm, respectively [18,19]. Moreover, halloysite is chemically similar to kaolinite and exhibits a predominantly hollow tubular structure with multiple siloxane (Si–O) groups located at the external surface [20]. Compared to MMT, there are only weak secondary interactions between adjacent tubes via hydrogen bonds and van der Waals forces and, therefore, no organomodification is habitually needed to achieve a homogeneous dispersion of HNTs in condensation polymers [21]. Besides, HNT exhibits high-cation exchange capability, thus it has high possibility to form hydrogen bonds by combination of their oxygen atoms with the hydrogen atoms from PLA [22]. 

Polymer nanocomposites containing HNTs have been prepared either via water solution mixing or melt blending due to they can be easily separated in polymer molten by shear based on their excellent hydrophilicity [23]. There are a few studies in the scientific literature reporting the incorporation of HNTs into PLA by different melt routes. For instance, Murariu et al. [24] reported a reinforcing phenomenon on PLA by using HNTs surface-treated by silanization with 3-(trimethoxysilyl) propyl methacrylate (TMSPM) and noticeable improvements in tensile strength were observed. Later, Liu et al. [25] prepared PLA/HNTs composites by melt compounding and injection molding. Hydrogen bonding interaction between PLA and HNTs was reported, showing a significant increase in tensile, flexural, and impact properties by nanocomposites prepared by melt compounding. PLA/HNTs composite foams were prepared by Wu et al. [26] by means of batch processing method using supercritical carbon dioxide (*Sc*-CO_2_) as physical blowing agent. The addition of 5 wt% HNTs increased the cell density and reduced the mean cell size. In another study, Prashantha et al. [27] incorporated quaternary ammonium salt with benzoalkonium chloride treated HNTs into PLA in a high shear twin-screw extruder using masterbatch dilution process, showing improvements in the mechanical properties due to the better interfacial compatibility induced by the modification of the nanotube surface. Stoclet et al. [28] compared the thermomechanical and flame retardant properties of PLA/HNTs nanocomposites prepared using a water-assisted extrusion process and conventional extrusion process. It was found that the injection of water into the polymer melt induced a better dispersion of the HNTs into the PLA matrix and also prevented polymer degradation during the extrusion due to a barrier effect of the water molecules that preferentially locate at the HNT–PLA interface.

However, the improvement in the properties is not only related to the nanofiller content but also to the dispersion of HNTs in the PLA and the interaction between HNTs and PLA. De Silva et al. [29] prepared PLA/HNTs films by solution casting method to ascertain their properties for packaging applications. It was observed that the films reinforced with 5 wt% of HNTs yielded the optimum results due to a better interfacial adhesion between HNTs and PLA, while at higher loadings the nanotubes highly aggregated and the mechanical properties were slightly inferior. Similarly, Othman et al. [30] analyzed the effect of incorporating different concentrations (0–5 wt%) of HNTs and glycerol on the mechanical and thermal properties of the nanocomposites. It was found that the addition of 3 wt% HNTs yielded the optimum mechanical properties due to the uniform distribution and dispersion of the nanoclays, while the addition of glycerol plasticized the nanocomposites to improve their processability. More recently, Risyon et al. [31] studied the effect of varying the nanoclays content on the properties of PLA films aimed to food packaging applications. The optimum concentration of HNTs was also 3 wt% and the resultant nanocomposites films demonstrated potential to extend the shelf life of packaged cherry tomatoes.

Being focused on the particular case of food packaging applications, this study reports whether the performance of PLA formulations can be tailored by the single and dual addition of OLA and HNTs. To this end, the HNTs were previously ultrasonicated in OLA liquid at mild temperature to improve their dispersion and the films were prepared by thermo-compression. The optical, morphological, thermal, mechanical, and barrier properties of the resultant binary and ternary systems of PLA/OLA and PLA/OLA/HNTs films were analyzed and related to their compositions.

## 2. Results and Discussion

### 2.1. Optical Properties of the PLA/OLA/HNTs Films

The appearance of the PLA films obtained by thermo-compression after the individual addition of OLA and combined addition of OLA with HNTs are gathered in Figure 1. A simple naked eye observation of these images reveals that the transparency and color characteristics of the PLA film were not affected by the addition of OLA. However, when HNTs were incorporated, the films became a little more opaque and slightly yellow, although they were still contact transparent. Table 1 gathers the *L*a*b** coordinate values and the color difference (*ΔE_ab_**) between the neat PLA film and the PLA/OLA blend films and PLA/OLA/HNTs composite films. In the case of the PLA/OLA films, all samples yielded very similar color parameters than the neat PLA film, showing slightly higher values for the color coordinate *b** (blue to yellow). In particular, *b** increased from −0.48, for the neat PLA film, up to –0.22, for the PLA blend film with 20 wt% of OLA, whereas the *L** (luminance) and *a** (green to red) were 30.24 and −0.3, respectively, thus showing no significant difference (*p* < 0.05). Thus, the PLA/OLA films were slightly yellowish but color variation remained unnoticeable (*ΔE_ab_** < 1), which is in agreement with previous optical studies about plasticized PLA materials by OLA [11].

However, the incorporation of HNTs progressively reduced lightness and slightly induced certain green and blue tonalities to the PLA films. In particular, for the PLA/OLA composite filled with 9 parts per hundred resin (*phr*) of HNTs, *L** increased to 38.19 whereas *a** and *b** respectively decreased to −1.27 and −2.31. In the case of the PLA/OLA composite film with the lowest nanofiller content, that is 3 *phr* of HNTs, *ΔE_ab_** was 1.65, so that only an experienced observer could notice the color difference (*ΔE_ab_** ≥ 1 and < 2). In general, the color differences increased for the PLA/OLA films with loadings of 6 *phr* and 9 *phr* of HNTs. In particular, the *ΔE_ab_** values were 5.15 (noticeable color difference, *ΔE_ab_**  ≥  3.5 and  <  5) and 7.68 (different colors, *ΔE_ab_** ≥ 5), respectively. The significant color change induced in the PLA/OLA films can be ascribed to the titanoferrous impurities naturally present in HNT [32]. In any case, the contact transparency was preserved and the resultant nanocomposite films can be of interest for food packaging applications [33,34].

### 2.2. Morphology of the PLA/OLA/HNTs Films

Figure 2 shows the transmission electron microscopy (TEM) micrographs of the HNTs after ultrasonication. Figure 2a shows the morphology of one isolated HNT in which it is possible to distinguish the tubular structure of the nanoclay. Measurements taken on this TEM micrograph revealed that a single HNT presents a wall of 17.3 nm that is formed by rolled aluminosilicate layers (see inset with zoomed-in TEM image within Figure 2a) with a tubular shape sizing 23.4 nm in diameter due to the differential specific volume between both sides of the layer and the lumen. The particular shape of HNTs can also be clearly seen in Figure 2b, showing that lumen diameters were comprised between 19.5–26.9 nm. As also observed in this TEM image, even after ultrasonic treatment in acetone using a low viscosity medium, HNTs still tended to form aggregates due to presence of –OH groups [18,19]. However, Rong et al. [35] reported that ultrasonication is effective for breaking HNT bundles, although the nanotube’s length can also be reduced at high power and long times. Therefore, HNTs can be easily dispersed in low viscosity polar solvents, but their dispersion in hydrophobic polymers with high melt viscosity is not straightforward.

Figure 3 gathers the field emission scanning electron microscopy (FESEM) images taken at 2000× of the cryo-fractured surfaces of the PLA films containing different OLA loadings. Figure 3a shows the morphology attained for the neat PLA film in which a smooth and homogeneous surface with some microcracks and large filaments can be observed. This type of fracture suggests that the film was brittle, although certain plastic deformation also occurred. One can observe in Figure 3b that the addition of 5 wt% of OLA did not modify the type of fracture, nevertheless, the presence of microcracks was lower and some plastic flow lines can be observed, which suggests increased ductile properties. However, contents of 10 wt% and 20 wt% of OLA, respectively shown in Figure 3c,d, yielded to a rough surface with multiple macrocracks and thin plastic filaments. As also observed in our previous study [36], the presence of OLA successfully inhibited microcrack formation and the cracks grew to a greater extent, thus suggesting that the film absorbed more energy and deformed more intensively during fracture. It should be noted that no sign of phase separation could be detected since PLA and OLA have the same chemical structure. In particular, PLA and OLA show very similar solubility parameters, that is, 19.5 (J/cm^3^)^1/2^ and 17.7 (J/cm^3^)^1/2^ for the biopolymer and its oligomer, respectively, indicating that they are miscible at high ratios [10]. Furthermore, no evaporation of the OLA plasticizer or vapor release from the blends was observed during film formation at high temperatures and pressures, indicating that their thermal stability was enough for melt processing.

Particle dispersion is a key factor when using nanoparticles. This problem is much more pronounced for highly hydrophilic nanoparticles such as HNTs, which tend to form aggregates by conventional melt blending. Figure 4 shows a comparative study performed by FESEM on the fracture surfaces of the PLA/OLA films filled with 20 wt% of OLA and 6 *phr* of HNTs with two dispersion procedures prior to melt compounding. The images corresponding to the nanocomposite films prepared using HNTs dispersed by conventional magnetic stirring in OLA are shown in Figure 4a,b. The second process consisted on ultrasonication in OLA heated at 100 °C, to reduce its viscosity, and the fracture surfaces of the resultant nanocomposite films are included in Figure 4c,d. As one can see, the conventional process with magnetic stirring did not allow good dispersion and micro-aggregates sizing approximately 18 µm × 7 µm were enclosed in the PLA/OLA matrix. Furthermore, a poor interaction between the aggregate and the surrounding PLA/OLA matrix can be observed due to the large particle size. As opposite, good nanoparticle dispersion was attained in the surface fractures of the nanocomposite films obtained with HNTs previously ultrasonicated.

The usefulness of ultrasonication in OLA medium for the HNTs dispersion in PLA was further assessed as a function of the nanoclay content. Figure 5 gathers the FESEM images taken at 5000× of the surface fractures of the PLA films containing 20 wt% of OLA and different HNTs loadings. The type of fracture surface observed in the PLA film, shown in Figure 5a, was kept for all the ternary system films combining OLA and HNTs. Efficient dispersion of HNTs can be seen in Figure 5b when 3 *phr* of HNTs was incorporated into the PLA/OLA matrix (see white arrows). The homogeneous dispersion of HNTs can be related to their good affinity with both PLA and OLA. High interactions between the biopolymer and nanoclay are known to take place by the formation of hydrogen bonds between the terminal –OH groups of PLA and the Si–OH groups of HNTs [21] and also between the oxygen atoms of the carboxyl (–COOH) group of PLA and the hydrogen atoms of the –OH groups present in HNTs [25]. Furthermore, the negative charges of the oxygen atoms on the HNTs surface can attract the positive charges of the hydrogen atoms on the PLA/OLA matrix [28], thus potentially forming new hydrogen bonds. In Figure 5c, corresponding to composite film filled with 6 *phr* of HNTs, a combination of domains with well-dispersed HNTs (white arrows) and some small HNTs aggregates (white ellipses) in the 0.5–1.0 µm range can be observed. It is worthy to note that, even though some aggregates were formed, the original aggregate size of HNTs, sizing 18 µm in length and 7 µm in width, was remarkably reduced after the ultrasound-assisted dispersion process. Finally, a greater number of large aggregates were noticed in the FESEM micrograph for the fracture surface of the PLA/OLA film filled with 9 *phr* of HNTs, which is included in Figure 5d. The dispersion of this nanocomposite film was also observed using a focused ion beam scanning electron microscopy (FIB-SEM) equipped with tridimensional (3D) tomography. The resultant Appendix A taken by electron microscopy tomography (EMT) also demonstrated that, in this nanocomposite sample, nanotubes were highly dispersed within the PLA/OLA matrix but large aggregates or bundles of several microns were also formed. Moreover, many little cavities or micro-voids can be observed in the FESEM image, which led to a porous-like surface and suggests that the dispersed HNTs did not break but pulled out of the matrix during cryo-fracture. This finding has already been described by De Silva et al. [37], who correlated nanotubes pullout to their poor interfacial adhesion with the PLA matrix. This observation was also clearly detectable when the nanofiller contents were above 5 wt%, regardless of the processing method and it also intensified for high HNTs loadings [29].

### 2.3. Mechanical Properties of the PLA/OLA/HNTs Films

Table 2 gathers the results of the tensile tests of the thermo-compressed films of neat PLA, PLA/OLA, and PLA/OLA/HNTs composites. In relation to the neat PLA, one can observe that the film showed mechanical properties of a hard but brittle material with values of tensile elastic modulus (*E_t_*) of 2846.3 MPa, tensile strength at yield (*σ_y_*) of 54.3 MPa, and elongation at break (*ε_b_*) of 3.52%. As one expected, the addition of OLA resulted in a plasticization of the PLA film, yielding a significant increment in the flexibility and a decrease in tensile strength. In particular, the *ε_b_* values gradually increased to 3.99%, 5.69%, and 6.14%, for OLA contents of 5, 10, and 20 wt%, respectively, while *σ_y_* was reduced to 53.5, 49.1, and 42.3 MPa. In the case of the PLA film containing 10 and 20 wt% of OLA, this represents an increase in ductility of nearly 62% and 75%, respectively. One can also observe that the *E_t_* values also decreased, which can be directly related to the combined decrease in rigidity and the increase in ductility, since the tensile modulus represents the stress (σ) to strain (*ε*) ratio in the linear (elastic) region. A similar effect was reported by Armentano et al. [11], for PLA blends with poly(3-hydroxybutyrate) (PHB) incorporating different amounts of OLA. At plasticizer contents of 20 and 30 wt%, *ε_b_* increased approximately by 57% and 164%, respectively. In another study, Burgos et al. [10] also showed that PLA/OLA blends at 20–25 wt% of plasticizer yielded up to 75-fold increases in ductility compared to neat PLA, thus suggesting that the type of OLA has high influence on the final ductile properties. The behavior observed was explained by the reduction of the macromolecular chains cohesion when the OLA plasticizer lubricates the PLA matrix. In this regard, Martin and Avérous [13] found that 20 wt% of OLA reduced *E_t_* by 63% whereas *ε_b_* increased up to 200%. However, our recent study reported that the use of impact-modifier type of OLA unexpectedly reduced the ductility of injection-molded pieces of PLA due to it rather acted as rubber-like filler in the PLA matrix [36]. Therefore, the different values attained can be mainly related to the OLA type since the here-used grade is mainly aimed to plasticize biopolyesters and it will be discussed later.

With regards to the PLA/OLA/HNTs composite films, one can observe that the incorporation of 3 and 6 *phr* of HNTs progressively increased the *E_t_* and *σ_y_* values whereas *ε_b_* significantly decreased. At the highest nanofiller content, that is, 9 *phr* of HNTs, all the mechanical parameters drastically decreased. Therefore, the highest mechanical performance was attained for the PLA/OLA film containing 3 *phr* of HNTs, reaching *E_t_*, *σ_y_*, and *ε_b_* values of 2854.1 MPa, 57.8 MPa, and 4.23%, respectively, and thus outperforming the neat PLA film. These findings point out that the mechanical properties of the PLA/OLA/HNTs nanocomposite films were affected by the nanofiller content, which can be related to the dispersion of the HNTs in the PLA/OLA matrix observed above during the morphological analysis. Similar findings were reported, for instance, by De Silva et al. [29] who found that *σ_y_* of solution-casted PLA films increased by 40% with increasing HNTs up to 5 wt% whereas contents of 10 wt% decreased the strength of the composites. In another studies, Othman et al. [30,38] showed that 3 wt% of HNTs resulted in the optimal mechanical properties due to the uniform distribution or dispersion of the nanoclay, whereas contents above 5 wt% led to agglomeration. Risyon et al. [31] also found that the optimum concentration of HNTs in PLA films was 3 wt% since it led to higher numbers of hydrogen bonds between PLA and HNTs compared to the films loaded with higher concentrations of HNTs, which provided higher mechanical resistance and rigidity but also restricted the mobility of the PLA chains and thus reduced ductility of the PLA films. Therefore, the properties attained herein are in agreement with previous studies, showing that the optimum loading content of HNT in PLA is 3 *phr*. However, it is also worth noting that the mechanical properties were also enhanced for a content of 6 *phr* of HNTs, which suggests that the presence of OLA could potentially favor the dispersion of slightly higher amount of nanofiller by plasticizing the PLA matrix.

### 2.4. Thermal Properties of the PLA/OLA/HNTs Films

Figure 6 shows the differential scanning calorimetry (DSC) thermograms collected during the second heating of the thermo-compressed PLA films. Table 3 gathers the average values of the main thermal transitions obtained from the DSC curves. The enthalpies associated with the cold crystallization and the melting processes are also reported in the table. In the 60–70 °C range, one can observe a step change in the base lines, which corresponds to the *T_g_* of PLA. This second-order thermal transition was located at 62.5 °C for neat PLA. The exothermic peaks located between 100 °C and 125 °C correspond to the cold crystallization temperature (*T_cc_*) of PLA. In the case of the neat PLA film, this peak showed a low intensity and it was located at 121.2 °C. Finally, in the thermal range of 140–160 °C, the crystalline PLA domains melted. One can observe in the thermogram of the neat PLA that the melting process occurred in a single endothermic peak, centered at 150.8 °C. In addition to the characteristic values of *T_g_*, *T_cc_*, and, *T_m_*, the enthalpies corresponding to the cold crystallization (*ΔH_cc_*) and melting (*ΔH_m_*) were collected from the DSC curves. The latter parameter was determined to ascertain the maximum degree of crystallinity (*χ_c max_*), resulting in a value of approximately 18% for the neat PLA film. This parameter gives more information about the effect of the additives on PLA since it does not consider the crystals formed during cold crystallization.

The addition of OLA produced a gradual and significant reduction of all the thermal values observed by DSC, that is, *T_g_*, *T_cc_*, and *T_m_*. The more intense reduction was attained for *T_g_*, due to the plasticizing effect of OLA, showing values of 58.7, 54.3, and 43.9 °C for the films containing 5, 10, and 20 wt%*,* respectively. Similar reductions have been reported in previous studies [12,13], indicating that the addition of 10–20 wt% of OLA to PLA induced a decreased in *T_g_* in the 20–40 °C range. Furthermore, the presence of a single *T_g_*, located between that of PLA and OLA, being the latter reported in the 10–30 °C range [39], suggests that the biopolymer and its oligomer were fully miscible at all the given composition studied here. The fact that PLA cold crystallized at lower temperature can also be related to the plasticization phenomenon of PLA provided by OLA since it is known to promote the motion of the PLA chains [36]. Other authors have ascribed this effect to a heterogeneous nucleation of the short length OLA molecules by which the cold crystallization process of PLA is favored [40]. In the thermogram of the PLA samples containing 20 wt% of OLA, one can observe the presence of two overlapped peaks. The first one appeared at nearly 139 °C and the second one at 150 °C. This double-peak phenomenon has been ascribed to the coexistence of two different crystal size populations [41] or, more frequently, to crystal reorganization upon melting in biopolyesters [42]. During this process, imperfect crystals melt at lower temperatures and the amorphous regions order into packed spherulites with thicker lamellar thicknesses that, thereafter, melt at higher temperatures. Moreover, broader melting peaks were observed, pointing out to the formation of more heterogeneous crystallites. However, it should be noticed that the *T_m_* value that corresponds to the higher melting peak was significantly not different than that attained in the neat PLA sample and the other OLA-containing PLA films, suggesting that similar crystallites were formed in all the materials. Furthermore, the *χ_c max_* values increased with the OLA content, which has been previously related to the inherent plasticizing effect of OLA [12,36]. Therefore, the presence of OLA in the PLA matrix hindered crystallization to some extent by impairing proper chain packing. However, higher amount of biopolyester was able to crystallize due to the enhanced mobility of the PLA chains by the lubricating effect of the oligomer. This result confirms both the high miscibility and interaction between the biopolyester and its oligomer and it opens up the possibility to develop highly crystalline PLA materials.

As opposite to OLA, one can observe that the addition of HNTs shifted the *T_g_* of the PLA/OLA blend to slightly higher values, in the 46–48 °C range. Moreover, the thermal parameters showed no significant differences (*p* > 0.05) for all the nanocomposites. This result is in agreement with the studies performed by Liu et al. [25] and Wu et al. [26], who indicated that the filler hinders the movement of the PLA chains. It further supports the delay observed in cold crystallization, which was reduced from nearly 102 °C, for the unfilled blend, to up to approximately 108 °C for the nanocomposite film containing 9 *phr* of HNTs. Moreover, whereas the melting profile of the blend nearly remained constant irrespective of the amount of nanoclay, showing *T_m_* values in the 140–150 °C range, the *χ_c max_* values decreased from over 35% to values in the range of 31–32%. While most studies have reported that HNTs act as nucleating agents for PLA [27,28,31], the present results indicate that the presence of OLA in the PLA matrix could prevent the crystal growth from the nanoclays surface. This finding also suggests that OLA could be preferentially located in the nanofiller-to-matrix interface due to its higher miscibility with PLA, which can potentially favor HNTs dispersion but also suppresses their nucleation.

Thermogravimetric analysis (TGA) was carried out to study the thermal stability of the PLA/OLA/HNTs blends. Figure 7 shows the TGA curves of the PLA/OLA/HNTs films, in which Figure 7a includes the evolution of the mass with temperature, while Figure 7b shows their respective first derivate thermogravimetric (DTG) curves. The thermal stability values extracted from the TGA curves are summarized in Table 4, gathering the temperature required for a loss of weight of 5% (*T_5%_*), which is representative for the onset of degradation, the maximum degradation rate temperature (*T_deg_*), and the amount of residual mass at 600 °C.

One can observe that neat PLA thermally degraded in a single step in the range from approximately 330 to 400 °C as previously reported by Agüero et al. [43]. In particular, the neat PLA film showed respective *T_5%_* and *T_deg_* values of 342.2 and 383 °C. The inset image within Figure 7a shows that the addition of OLA gradually reduced the thermal stability of PLA, being more intensely for the onset of degradation since the *T_5%_* values gradually reduced to 335.3, 313.3, and 292.5 °C for additions of 10, 20, and 30 wt%, respectively. This is in agreement with previous studies reporting about the thermal stability of PLA/OLA blends [12,36,44], which related the lower stability of the blends to the lower *T_g_* and the inherently poor thermal stability of OLA. Finally, all the PLA/OLA blends showed residual mass values below 2 wt%.

On one hand, the addition of 3 and 6 *phr* of HNTs, particularly the lowest content, induced a positive increase in the onset degradation, showing *T_5%_* values of 299.5 and 296.5 °C, respectively. On the other hand, HNTs contents of 9 *phr* reduced the thermal stability of the PLA/OLA film, whereby both *T_5%_* and *T_deg_* decreased by approximately 10 °C and 33 °C, respectively. These results indicate that the addition of low HNTs amounts successfully improved the thermal stability of PLA/OLA film due to the good dispersion of HNTs in the PLA matrix. This finding is in agreement with the study recently performed by Risyon et al. [31], who found that *T_5%_*, *T_deg_*, and residual mass improved since well-dispersed HNTs formed tortuous paths that provided physical thermal barriers to the PLA film. In this regard, nanoclays can be responsible for delaying the sample weight loss by the formation of tortuous paths that hinders the diffusion of volatile degradation products out of the material and, for the particular case of HNTs, certain sorption during degradation could additionally occur due to their hollow structure and high porosity [45]. The fact that thermal stability was lower at the highest HNTs content further confirms that the nanofillers agglomerated and, hence, reduced the heat barrier effect and catalyzed thermal degradation. The catalytic action of halloysite on the pyrolysis of PLA has been ascribed to the presence of Brönsted acid sites, such as hydrophilic Si–OH and aluminum hydroxide (Al–OH) groups, on the external surface of the clay [28]. One can also observe that the residual mass progressively increased with the HNT content since the nanoclay is thermally stable at high temperatures and it can also act as a heat barrier, which could enhance the formation of char after thermal decomposition.

### 2.5. Thermomechanical Properties of the PLA/OLA/HNTs Films

Dynamic mechanical thermal analysis (DMTA) was carried out to determine the thermomechanical properties of the PLA/OLA/HNTs films. In Figure 8 the storage modulus (*E’*) and dynamic damping factor (*tan δ*) were recorded as a function of temperature. Figure 8a gathers the evolution of *E’* with temperature in which one can observe that all the PLA-based films presented a similar profile. In particular, the samples showed high *E’* values, that is, high stiffness, at temperatures below 40 °C and then *E’* sharply decreased from 50 to 80 °C. This thermomechanical change indicates that the alpha (α)-relaxation of the biopolymer, which is related to its *T_g_*, was exceeded. Finally, *E’* increased in the 85–100 °C range due to the occurrence of cold crystallization. These results correlated well with the DSC analysis described above.

Table 5 shows the *E’* values of the storage modulus measured at 30, 75, and 110 °C, being these temperatures representative of the stored elastic energy of PLA in its three states, that is, amorphous glassy, amorphous rubber, and semi-crystalline. The addition of OLA shifted the *E’* curves to significantly lower temperatures, confirming the plasticizing effect of the oligomer, which is in agreement with both the DSC analysis and our previous study [36]. It is worthy to note, however, that the *E’* values increased, especially at the 10 wt% loading. This observation suggests that the OLA-containing PLA films developed higher crystallinity, which was also observed during the DSC analysis. This increment in rigidity was more noticeable at 110 °C, once PLA finished its cold crystallization. In particular, the *E’* values increased from 30.1 MPa, for the neat PLA film, up to 93.2 MPa, for the PLA film containing 10 wt% of OLA. One can also observe that lower values were attained at 20 wt% of OLA, which can be related to the formation of a highly plasticized PLA structure. As determined above during the mechanical properties, the addition of HNTs in the glassy state yielded an increase in *E’*. Similar results were reported by Prashantha et al. [27], who observed a general increase of *E′* with increasing the nanofiller content. However, at high temperatures, the nanocomposite films showed lower *E’* values than the unfilled PLA/OLA films, which can also be ascribed to the potential of the latter sample to develop higher crystallinity. One can also notice that the PLA/OLA films with the lowest HNTs contents, that is, 3 *phr* and 6 *phr*, showed significantly higher values than that containing 9 *phr* due to the higher dispersion attained in these nanocomposite films.

Figure 8b shows the evolution of *tan δ* versus temperature. The *tan δ* peaks represent the change in the thermomechanical behavior of PLA when the α-relaxation of the biopolymer is reached, which relates to the *T_g_* of the biopolyester. As also shown in Table 5, the α-relaxation peak for the neat PLA film was located at 64.1 °C, which is slightly lower than the *T_g_* value obtained by DSC analysis. One can observe that the peak values gradually decreased with the OLA content, down to 52 °C for the PLA film containing 20 wt%, being ascribed to the plasticization of the PLA matrix as also stated by DSC. The *tan δ* peaks of PLA/HNTs nanocomposite films slightly broaden and shifted to lower temperatures compared to that of neat PLA for contents of 6 *phr* and 9 *phr* of HNTs. Broadening of *tan δ* and eventual shift to lower temperatures for nanocomposites indicate an increase in the segmental motions of the polymer blend [46]. This result observed for the nanocomposite films with high nanoclay loadings differs with the above-shown DSC results and also previous DMTA studies of PLA/HNTs composites [27]. The here-attained thermomechanical change suggests a hydrolytic effect of the nanoclay during extrusion on the PLA/OLA blend by which the biopolymer’s molecular weight (*M_W_*) is reduced and the molecular motion is favored due the chains are shorter.

### 2.6. Barrier Properties of the PLA/OLA/HNTs Films

The water vapor permeability (WVP) and limonene permeability (LP) of the PLA films containing OLA and HNTs are shown in Table 6. Biodegradable films for packaging applications habitually show low water barrier resistance, which strongly limits their application for food packaging purposes due to physical and chemical deterioration of foodstuff. It can be observed that the neat PLA film presented a WPV value of 1.22 × 10^−14^ kg·m/m^2^·Pa^1^·s^1^, which indicates that PLA is a medium-to-low barrier material to water vapor, being similar to that reported in our previous study [9]. Unexpectedly, the addition of OLA reduced the permeability to water vapor of the PLA film. Particularly, the blend containing 10 wt% of OLA showed the highest barrier properties with a WVP value of 0.65 × 10^−14^ kg·m/m^2^·Pa^1^·s^1^. This barrier enhancement can be attributed to the densification of PLA produced by the increase in crystallinity after the OLA addition [47], which was also previously corroborated by DSC and DMTA. Similar results were reported by Ambrosio-Martín et al. [44], who associated the reduction of the vapor permeability with an effect of “antiplasticization”. In this regard, several works have demonstrated reductions in gas permeability associated with a fractional free volume reduction phenomenon in polymers by the incorporation of low amounts of low-*M_W_* additives [48,49]. Since water is a condensable molecule and hence permeability is based on a solubility phenomenon rather than diffusion one, the reduced concentration of sorbed water vapor due to the occupancy of free volume of the amorphous regions PLA by the OLA molecules could explain the higher barrier properties. The higher value observed for the PLA blend with 20 wt% of OLA further confirms the highly plasticization attained in this PLA film, which seems to predominate over “antiplasticization” at high contents. In any case, all the OLA-containing PLA films showed significantly higher barrier performance than the neat PLA film. Similar results were obtained during the analysis of the limonene transport properties, which are usually carried out as a standard system to test aroma barrier. The LP value was reduced from 3.33 × 10^−15^ kg·m/m^2^·Pa^1^·s^1^, for the neat PLA film, to 0.94 × 10^−15^ kg·m/m^2^·Pa^1^·s^1^ and 1.72 × 10^−15^ kg·m/m^2^·Pa^1^·s^1^ for the PLA films containing 10 and 20 wt% of OLA, respectively. Therefore, the here-studied type of OLA performed as an “antiplasticizer” at contents of 10 wt% in the PLA, whereas higher concentrations trigger plasticization possibly through the formation of clusters of additive molecules. These results confirm that the addition of low-to-medium contents of OLA is very promising for the development of medium barrier films of PLA.

Interestingly, the incorporation of low amounts of HNTs induced a further significant reduction of both water and limonene vapors. In particular, the PLA/OLA nanocomposite film containing 3 *phr* of HNTs showed WPV and LP values of 0.80 × 10^−14^ kg·m/m^2^·Pa^1^·s^1^ and 0.91 × 10^−15^ kg·m/m^2^·Pa^1^·s^1^, respectively, which represents reductions of approximately 8% and 47% compared with the unfilled PLA/OLA film. This barrier enhancement can be ascribed to the efficient dispersion of HNTs in the PLA/OLA matrix, which results in the formation of tortuous paths that reduced the diffusion of the vapor molecules through the PLA film [50]. The lower reduction attained for water vapor in comparison with limonene vapor can related to the hydrophilic nature of HNTs since they exhibit permanent negative charges on their surface by which water molecules entrapment can be increased [51]. As similar to other properties, when higher contents of HNTs were incorporated, that is, 6 and 9 *phr*, the WVP and LP values increased, which can also be due to agglomeration of nanoclays by which no tortuous path was formed. Similar results were reported by Risyon et al. [31], who showed water and oxygen barrier improvements of approximately 21% and 33%, respectively, when 3 *phr* of HNTs was incorporated into PLA films whereas higher contents led to higher permeability.

## 3. Materials and Methods

### 3.1. Materials

PLA Ingeo^TM^ Biopolymer 2003D was supplied in pellets form by NatureWorks LLC (Minnetonka, MN, USA). This PLA grade has a melt flow rate (MFR) of 6 g/10 min measured at 210 °C and 2.16 kg and a true density of 1.24 g/cm^3^. It has been specifically designed for extrusion for use in fresh food packaging and food serviceware applications. OLA was provided by Condensia Química S.A. (Barcelona, Spain) as Glyplast^®^ OLA 8 in liquid form. It has a density of 1.11 g/cm^3^, a viscosity of 22.5 mPa·s at 100 °C, an ester content >99%, a maximum acid index of 2.5 mg KOH/g, and a maximum water content of 0.1 wt%. According to the manufacturer, this OLA grade is obtained from renewable raw materials and is biodegradable (UN 20200:2006), being specifically designed as PLA plasticizer to obtain stretch films. HNTs were purchased at Sigma-Aldrich S.A. (Madrid, Spain) with reference 685445 in powder form. HNTs are characterized by a *M_W_* of 294.19 g/mol and a true density of 2.53 g/cm^3^. According to the manufacturer, the nanotubes are 1–3 μm long with a diameter in the 30–70 nm range, resulting in a surface area of 64 m^2^/g, a pore volume of 1.26–1.34 mL/g, and a cation exchange capacity of 8.0 meq/g. Acetone and _D_-limonene, with 98% purity, were also obtained from Sigma-Aldrich S.A.

### 3.2. Films Preparation

The as-received PLA pellets were dried overnight in a dehumidifier MDEO from Industrial Marsé (Barcelona, Spain) at 60 °C to eliminate residual moisture since the biopolyester has a high sensitivity to hydrolysis. The OLA content was tested in concentrations of 5, 10, and 20 wt% with respect to PLA. Thereafter, for a fixed a content of 20 wt% of OLA, HNTs were incorporated at 3, 6, and 9 *phr* of PLA/OLA blend. The selected content of OLA was chosen since it yielded the highest stability and performance for PLA according to Burgos et al. [10]. The corresponding HNTs loadings were previously dispersed in OLA by the ultrasound-assisted technique in an ultrasonicator model Sonoplus HD 2200 from Bandelin Electronic GmbH & Co. KG (Berlin, Germany) for 5 min using an amplitude of 80% to attain homogenous mixtures. As the OLA additive is a rather viscous liquid at room temperature, it was first heated up to 60 °C. During the ultrasound assisted dispersion the temperature increased up to 100 °C due to internal friction. Conventional magnetic stirring for 5 min was also performed for each HNTs-containing OLA dispersion min for comparison. Table 7 gathers the code and composition of the prepared samples.

The mixtures were melt-compounded in a twin-screw extruder from Construcciones Mecánicas Dupra, S.L. (Alicante, Spain) equipped with a screw diameter of 25 mm and a length-to-diameter (L/D) ratio of 24. Further details of the equipment can be found elsewhere [52]. The rotating speed was set to 25 rpm and the temperature profile was: 165 °C (feeding hopper)–170 °C–175 °C–180 °C (extrusion die). The strands were cooled in air, granulated in pellets in an air-knife unit, and dried at 60 °C for 72 h to remove moisture. Thereafter, the pellets were thermo-compressed using a 10-Tn hydraulic press from Robima S.A. (Valencia, Spain) equipped with two hot aluminum plates and a temperature controller from Dupra S.A. (Castalla, Spain) [53]. For this, about 5 g of each composition was thermo-compressed at 180 °C with a pressure of 40 MPa tons for 3 min in a square frame of 10 cm × 10 cm by two hot plates and air cooled at room conditions. Films with a mean thickness of approximately 250 µm were attained and stored at 23 °C y 50% relative humidity (HR) for, at least, 15 days prior to characterization.

### 3.3. Films Characterization

#### 3.3.1. Color Measurements

Changes in color were measured in a colorimetric spectrophotometer Konica CM-3600d Colorflex-DIFF2, from Hunter Associates Laboratory, Inc. (Reston, VA, USA). The Commission Internationale de l’Eclairage (CIE) standard illuminant D65 was used to measure the CIE Lab color space coordinates *L*a*b** using an observer angle of 10°. In the selected *L*a*b** color space, *L** stands for the luminance, where *L** = 0 represents dark and *L** = 100 indicates clarity or lightness, and the *a*b** pair represents the chromaticity coordinate, where *a** > 0 is red, *a** < 0 is green, *b** > 0 is yellow, and *b** < 0 is blue. The *L*a*b** coordinate values were obtained on five different film samples and the color change, that is, *ΔE_ab_**, was calculated using Equation (1):(1)$$\Delta E^{*}{}_{a,b}=\sqrt{{\left({\Delta L}^{*}\right)}^{2}+{\left({\Delta a}^{*}\right)}^{2}+{\left(\Delta b^{*}\right)}^{2}}$$
where *ΔL**, *Δa**, and *Δb** correspond to the differences between the color parameters of the tested films and the values of the neat PLA film. Color change evaluation was performed using the following assessment [54]: unnoticeable (*ΔE_ab_** < 1), only an experienced observer can notice the difference (*ΔE_ab_** ≥ 1 and < 2), an inexperienced observer notices the difference (*ΔE_a_** ≥ 2 and < 3.5), the samples show clear noticeable difference (*ΔE_ab_** ≥ 3.5 and < 5), and the observer notices different colors (*ΔE_ab_** ≥ 5).

#### 3.3.2. Microscopy

The morphology of HNTs was studied by TEM using a Philips CM10 (Eindhoven, the Netherlands) with an acceleration voltage of 100 kV. Prior to TEM observation, a small amount of HNTs was dispersed in acetone and subjected to ultrasound dispersion in an ultrasound bath Sonicador SONOPULS from Bandelin Electronic GmbH & Co. KG for 5 min at room temperature. Then, a drop of this dispersion was poured onto a carbon grid and subjected to solvent evaporation at room temperature.

The films were cryo-fractured by immersion in liquid nitrogen and their fracture surfaces were observed by FESEM in a ZEISS ULTRA 55 FESEM microscope from Oxfrod Instruments (Abingdon, UK). The microscope worked at an acceleration voltage of 2 kV. The fracture surfaces were previously coated with a gold-palladium alloy in a Quorun Technologies Ltd. EMITECH mod. SC7620 sputter coater (East Sussex, UK).

#### 3.3.3. Mechanical Tests

The mechanical properties of the films were determined by tensile tests in a universal test machine ELIB 30 from S.A.E. Ibertest (Madrid, Spain) following the guidelines of ISO 527-3:2018 with rectangular samples sizing 100 mm × 10 mm. The selected load cell was 10 kN whereas the cross-head speed was set to 2 mm/min. At least six different specimens per formulation were tested in room conditions.

#### 3.3.4. Thermal Tests

DSC was carried out to study the thermal transitions of samples in a Mettler-Toledo 821 calorimeter (Schwerzenbach, Switzerland). An average weight of 5–7 mg of each sample was placed in 40-µL aluminum-sealed crucible and subjected to a three-step program in inert atmosphere of nitrogen with a constant flow of 30 mL/min. The samples were heated from 30 to 200 °C, cooled down to 0 °C, and heated up again to 250 °C. All heating/cooling rates were set at 10 °C/min and *χ_c max_* was calculated using Equation (2) [36]:(2)$$X_{\mathrm{Cmax}}=\left[\frac{\Delta H_{m}}{\Delta H_{m}^{0}\cdot \left(1-w\right)}\right]\;\cdot \;100\;\left(\%\right)$$
where *∆**H*_m_ (J/g) corresponds to the melting enthalpy of PLA, ∆*H*_m_^0^ (J/g) is the theoretical value of a fully crystalline PLA, that is, 93.0 J/g [55], and 1-*w* indicates the weight fraction of PLA in the sample.

TGA was performed to determine the thermal stability of films using a TGA/SDTA 851 thermobalance from Mettler-Toledo Inc. (Schwerzenbach, Switzerland). Samples with an average weight between 5 and 7 mg were placed in standard alumina crucibles (70 μL) and subjected to a single-step thermal program from 30 to 650 °C at a heating rate of 20 °C/min in an air atmosphere. All the thermal tests were performed by triplicate.

#### 3.3.5. Thermomechanical Tests

DMTA was conducted in a DMA-1 model from Mettler-Toledo S.A. (Barcelona, Spain), working in tension mode. Rectangular film samples sizing 10 mm × 5 mm were subjected to a temperature sweep program from −50 °C to 120 °C at a heating rate of 2 °C/min. The offset strength was set at 1 N, the offset deformation at 150%, and the control deformation at 6 µm. The DMTA tests were run in triplicate to obtain reliable data.

#### 3.3.6. Permeability Measurements

WVP was determined for the PLA-based films according to the ASTM E96-95 gravimetric method. For this, 5 ml of distilled water were poured into a Payne permeability cup (∅ = 3.5 cm) from Elcometer Sprl (Hermallesous-Argenteau, Belgium). The films were placed in the cups so that on one side they were exposed to 100% RH, avoiding direct film contact with water. The cups containing the films were then secured with silicon rings and stored in a desiccator at 25 °C and 0% RH. Identical cups with aluminum foils were used as control samples to estimate water loss through the sealing. The cups were weighed periodically using an analytical balance with ± 0.0001 g accuracy. Water vapor permeation rate (WVPR), also called water permeance when corrected for permeant partial pressure, was determined from the steady-state permeation slope obtained from the regression analysis of weight loss data per unit area versus time, in which the weight loss was calculated as the total cell loss minus the loss through the sealing. WVP was obtained, in triplicate, by correcting the permeance by the average film thicknesses.

LP was also determined according to ASTM E96-95 gravimetric method. Similarly, 5 ml of _D_-limonene was placed inside the Payne permeability cups and the cups containing the films were stored under controlled conditions, that is, 25 °C and 40% RH. Limonene permeation rate (LPR) was obtained from the steady-state permeation slopes. The weight loss was calculated as the total cell loss minus the loss through the sealing plus the water sorption gained from the environment measured in samples with no permeant. LP was calculated taking into account the average sheet thickness in each case, measuring three replicates per sample.

### 3.4. Statistical Analysis

The values of the film properties were evaluated by analysis of variance (ANOVA) with 95% confidence interval level (*p* ≤ 0.05). For this purpose, a multiple comparison test (Tukey) was followed using the software OriginPro8 (OriginLab Corporation, Northampton, MA, USA).

## 4. Conclusions

Due to their renewable origin and high ductility, OLA-containing PLA films have recently received considerable attention for a use in compostable flexible packaging. The addition of the here-tested OLA plasticizer did not only contribute to the improvement of the ductility and reduction of PLA’s *T_g_* but also, up to contents of 10 wt%, it performed as an “antiplasticizer”, making the films more crystalline and attractive for medium barrier applications. However, despite these advantages, the resultant blend films also showed some limitations, such as lower mechanical strength and reduced thermal stability. To optimize the properties of the PLA/OLA blend films, different amounts of HNTs were co-added with OLA by means of ultrasonication at mild temperature to facilitate their subsequent dispersion in the PLA matrix. According to optical, mechanical, thermal, and barrier characterizations, the performance of the plasticized PLA films was significantly improved by reinforcement of the nanofillers. It was observed that either the individual addition of OLA or the combined addition with low HNTs loadings did not significantly alter the transparency and color properties of the PLA films. The morphological analysis revealed that the use of ultrasound-assisted HNTs dispersion significantly avoided formation of aggregates in the PLA/OLA matrix, particularly for films with the lowest HNTs contents, that is, 3 *phr*. Nevertheless, higher HNTs loadings resulted in the formation of small aggregates that led to the drop of the mechanical and thermal properties of the blend and also reduced lightness and induced color changes to the PLA films.

From the above, the present research study demonstrates that the mechanical and thermal properties of PLA films can be tuned by carefully selecting the contents added of OLA plasticizer and HNTs, while improving other valuable properties such as the thermomechanical resistance and barrier against water and aroma vapors. One can, therefore, conclude that PLA films with low-to-medium contents of OLA and low loadings of HNTs are technically viable for use in applications in the food packaging industry.

## Figures and Tables

**Figure 1 molecules-25-01976-f001:**
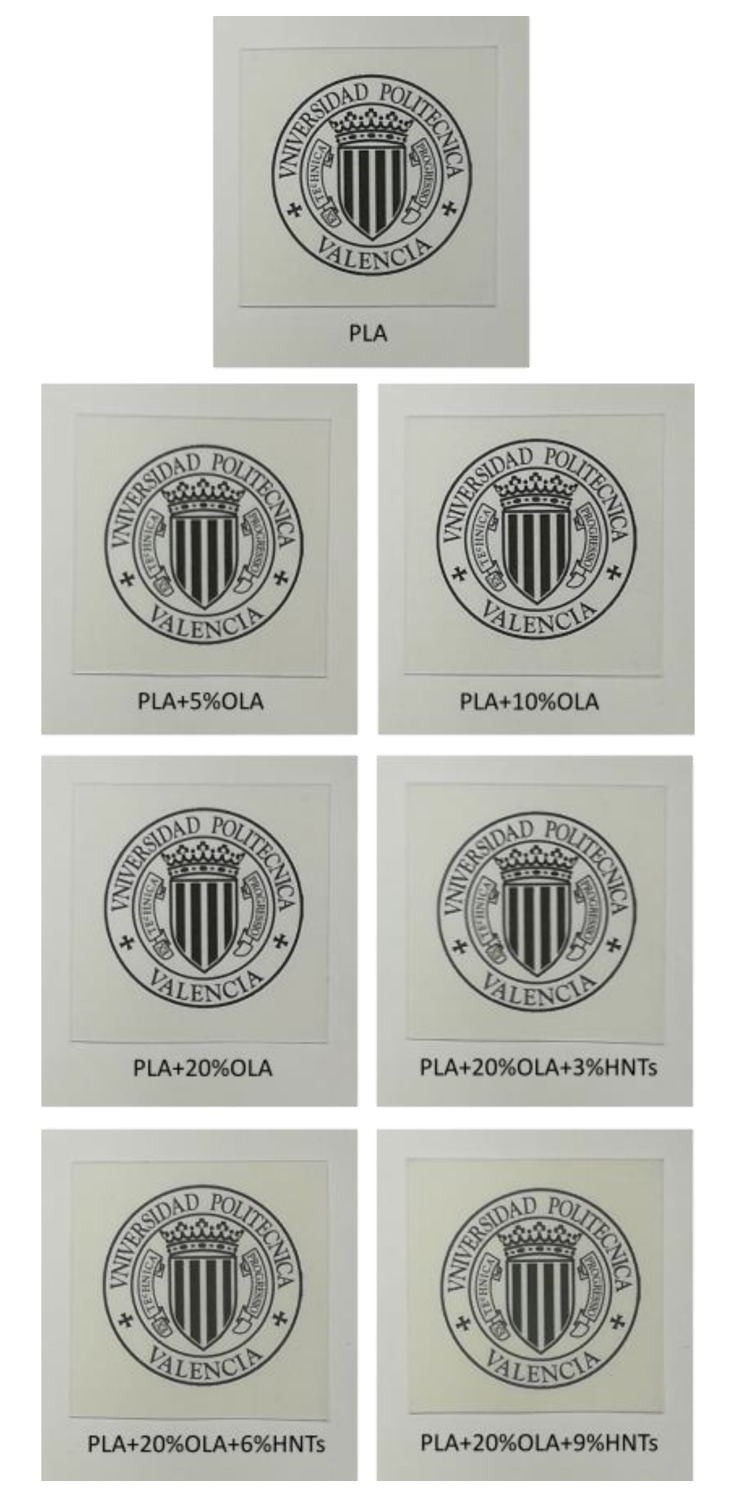
Visual aspect of the polylactide (PLA)/oligomer of lactic acid (OLA)/halloysite nanotubes (HNTs) films.

**Figure 2 molecules-25-01976-f002:**
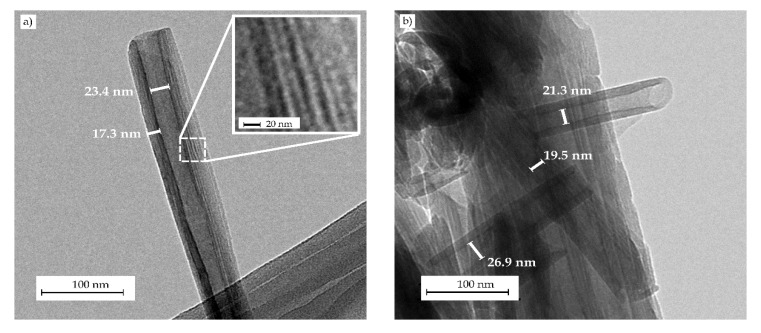
Transmission electron microscopy (TEM) images corresponding to (**a**) individual isolated halloysite nanotube (HNT) and (**b**) aggregate of HNTs. Images were taken at 52,000× showing scale markers of 100 nm.

**Figure 3 molecules-25-01976-f003:**
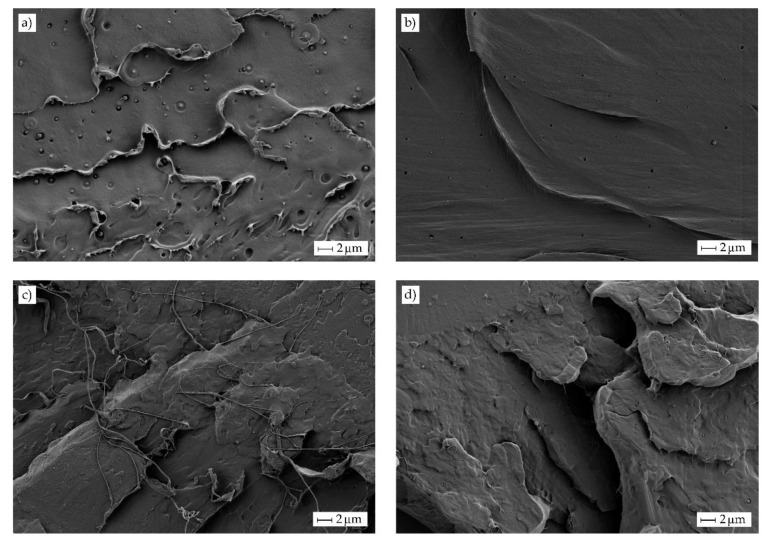
Field emission scanning electron microscopy (FESEM) images of the fracture surfaces of the polylactide (PLA) films with different weight contents (wt%) of oligomer of lactic acid (OLA): (**a**) PLA; (**b**) PLA + 5 wt% OLA; (**c**) PLA + 10 wt% OLA; (**d**) PLA + 20 wt% OLA. Images were taken at 2000× showing scale markers of *2* µm.

**Figure 4 molecules-25-01976-f004:**
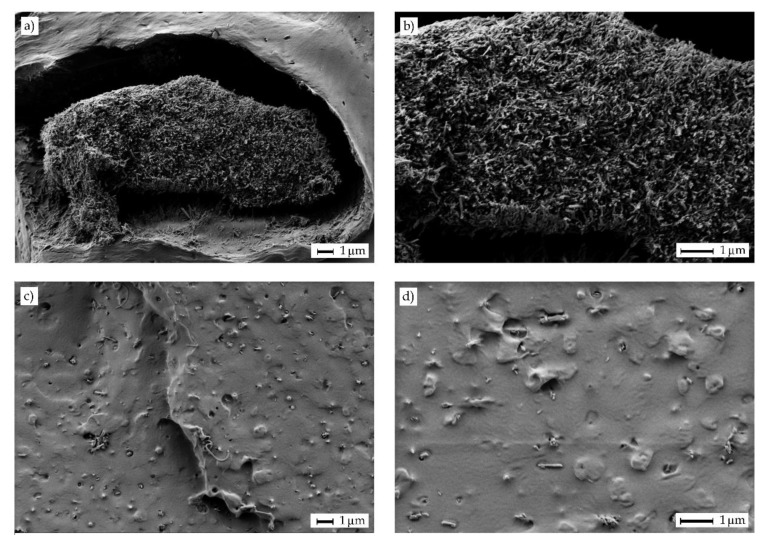
Field emission scanning electron microscopy (FESEM) images of the fracture surfaces of the polylactide (PLA) films containing 20 wt% of oligomer of lactic acid (OLA) and 6 parts per hundred resin (*phr*) of halloysite nanotubes (HNTs): (**a,b**) HNTs were dispersed in OLA by magnetic stirring; (**c,d**) HNTs were ultrasonicated in OLA at 100 °C. Images on the left were taken at 5000×, while images on the right at 10000×. Scale markers at both magnifications represent 1 µm.

**Figure 5 molecules-25-01976-f005:**
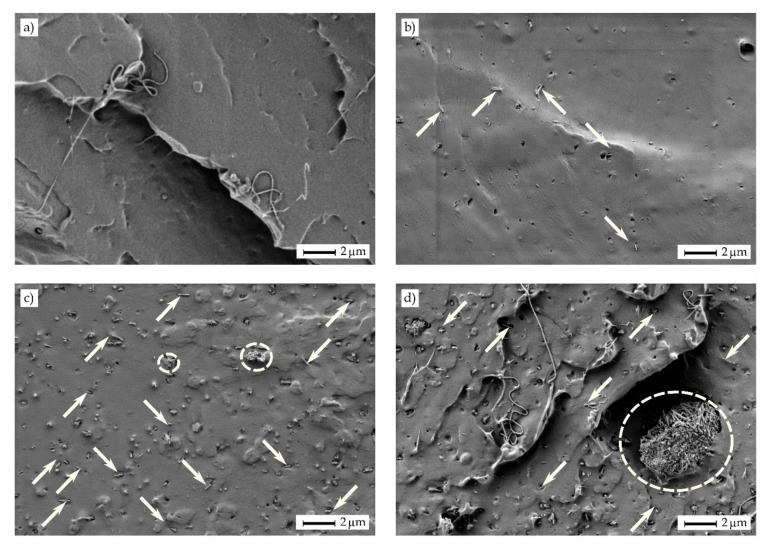
Field emission scanning electron microscopy (FESEM) images of the fracture surfaces of the polylactide (PLA) films containing 20 wt% of oligomer of lactic acid (OLA) and different parts per hundred resin (*phr*) of halloysite nanotubes (HNTs): (**a**) PLA; (**b**) PLA + 20 wt% OLA + 3 *phr* HNTs; (**c**) PLA + 20 wt% OLA + 6 *phr* HNTs; (**d**) PLA + 20 wt% OLA + 9 *phr* HNTs. Images were taken at 5000× with scale markers of *2* µm.

**Figure 6 molecules-25-01976-f006:**
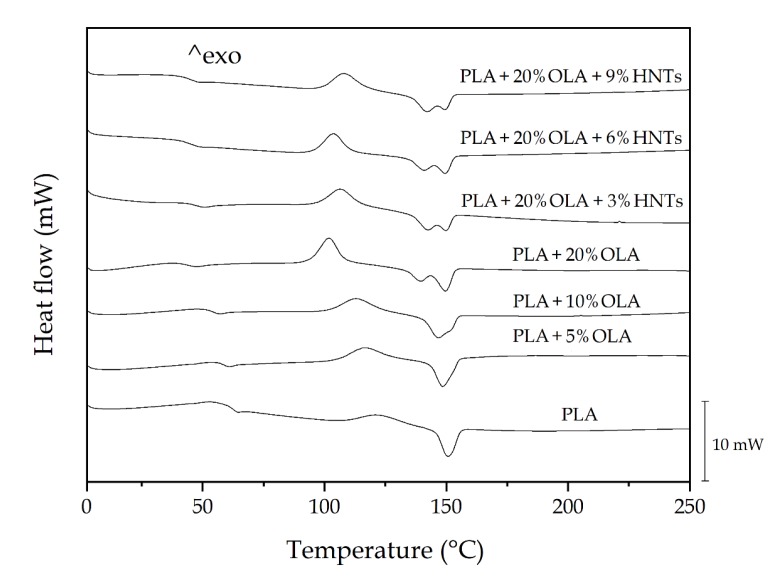
Differential scanning calorimetry (DSC) thermograms taken during second heating of the polylactide (PLA)/oligomer of lactic acid (OLA)/halloysite nanotubes (HNTs) films.

**Figure 7 molecules-25-01976-f007:**
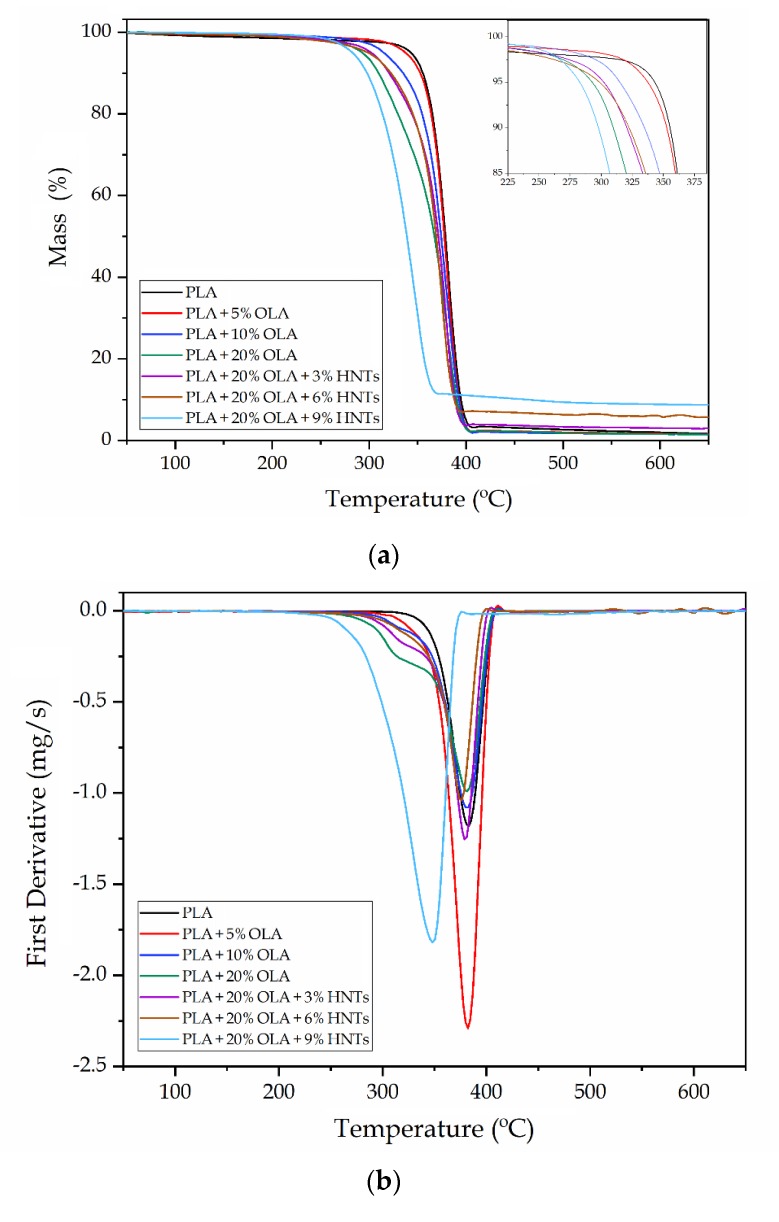
(**a**) Thermogravimetric analysis (TGA) and (**b**) first derivate thermogravimetric (DTG) curves of the polylactide (PLA)/oligomer of lactic acid (OLA)/halloysite nanotubes (HNTs) films.

**Figure 8 molecules-25-01976-f008:**
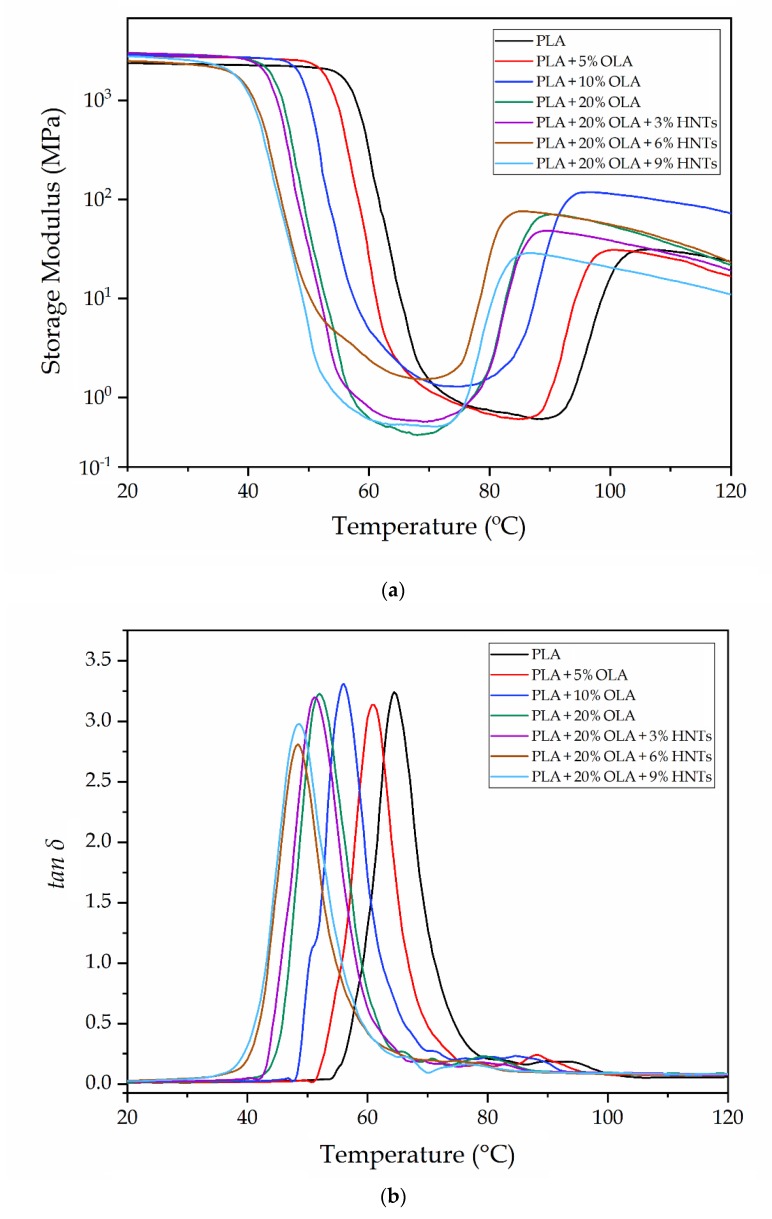
Evolution as a function of temperature of the (**a**) storage modulus and (**b**) dynamic damping factor (*tan δ*) of the polylactide (PLA)/oligomer of lactic acid (OLA)/ halloysite nanotubes (HNTs) films.

**Table 1 molecules-25-01976-t001:** Color parameters (*L**, *a**, *b**) and color difference (*ΔE_ab_****) of the polylactide (PLA)/oligomer of lactic acid (OLA)/ halloysite nanotubes (HNTs) films.

Film	*L**	*a**	*b**	*ΔE_a,b_**
PLA	30.79 ± 0.31 ^a^	−0.32 ± 0.06 ^a^	−0.48 ± 0.03 ^a^	-
PLA + 5% OLA	30.67 ± 0.27 ^a^	−0.38 ± 0.04 ^a^	−0.30 ± 0.09 ^b^	0.36 ± 0.03 ^a^
PLA + 10% OLA	30.36 ± 0.15 ^a^	−0.25 ± 0.05 ^a^	−0.15 ± 0.08 ^c^	0.64 ± 0.09 ^b^
PLA + 20% OLA	30.24 ± 0.03 ^a^	−0.30 ± 0.06 ^a^	−0.22 ± 0.08 ^bc^	0.66 ± 0.08 ^b^
PLA + 20% OLA + 3% HNTs	32.35 ± 0.06 ^b^	−0.61 ± 0.04 ^b^	−0.89 ± 0.05 ^d^	1.65 ± 0.25 ^c^
PLA + 20% OLA + 6% HNTs	35.72 ± 0.27 ^c^	−1.01 ± 0.02 ^c^	−1.76 ± 0.11 ^e^	5.15 ± 0.48 ^d^
PLA + 20% OLA + 9% HNTs	38.19 ± 0.64 ^d^	−1.27 ± 0.13 ^d^	−2.31 ± 0.15 ^f^	7.68 ± 0.39 ^e^

^a–f^ Different letters in the same column indicate a significant difference among the samples (*p* < 0.05).

**Table 2 molecules-25-01976-t002:** Mechanical properties of the polylactide (PLA)/oligomer of lactic acid (OLA)/halloysite nanotubes (HNTs) films in terms of elastic modulus (*E_t_*), strength at yield (*σ_y_*), and elongation at break (*ε_b_*).

Film	*E_t_* (MPa)	*σ_y_* (MPa)	*ε_b_* (%)
PLA	2846.3 ± 137 ^a^	54.3 ± 0.9 ^a^	3.52 ± 0.2 ^a^
PLA + 5% OLA	2745.6 ± 146 ^a^	53.5 ± 1.0 ^a^	3.99 ± 0.5 ^a^
PLA + 10% OLA	2532.8 ± 103 ^b^	49.1 ± 0.7 ^b^	5.69 ± 0.3 ^b^
PLA + 20% OLA	2469.4 ± 112 ^b^	42.3 ± 0.9 ^c^	6.14 ± 0.3 ^b^
PLA + 20% OLA + 3% HNTs	2854.1 ± 103 ^a^	57.8 ± 1.0 ^d^	4.23 ± 0.2 ^c^
PLA + 20% OLA + 6% HNTs	2927.6 ± 105 ^a^	47.8 ± 1.1 ^b^	3.74 ± 0.4 ^a^
PLA + 20% OLA + 9% HNTs	2651.3 ± 102 ^ab^	39.1 ± 1.3 ^e^	3.16 ± 0.6 ^d^

^a–e^ Different letters in the same column indicate a significant difference among the samples (*p* < 0.05).

**Table 3 molecules-25-01976-t003:** Thermal properties of the polylactide (PLA)/oligomer of lactic acid (OLA)/halloysite nanotubes (HNTs) films in terms of glass transition temperature (*T_g_*), cold crystallization temperature (*T_cc_*), melting temperature (*T_m_*), cold crystallization enthalpy (∆*H_cc_*), melting enthalpy (∆*H_m_*), and degree of crystallinity (χ_c max_).

Film	*T_g_* (°C)	*T_cc_* (°C)	*T_m_* (°C)	*Δ**H_cc_* (J/g)	*Δ**H_m_* (J/g)	*χ_c max_* (%)
PLA	62.5 ± 0.6 ^a^	121.2 ± 0.8 ^a^	150.8 ± 0.5 ^a^	6.98 ± 0.8 ^a^	16.80 ± 0.5 ^a^	18.06 ± 0.51 ^a^
PLA + 5% OLA	58.7 ± 0.5 ^b^	116.1 ± 0.4 ^b^	148.7 ± 0.3 ^b^	15.92 ± 0.4 ^b^	20.68 ± 0.3 ^b^	23.41 ± 0.32 ^b^
PLA + 10% OLA	54.3 ± 0.6 ^c^	112.8 ± 0.7 ^c^	146.9 ± 0.4 ^c^	17.03 ± 0.4 ^c^	21.63 ± 0.4 ^c^	25.84 ± 0.54 ^c^
PLA + 20% OLA	43.9 ± 0.8 ^d^	101.9 ± 0.6 ^d^	138.8 ± 0.6^d^ / 150.0 ± 0.5 ^a^	25.63 ± 0.6 ^d^	26.20 ± 0.6 ^d^	35.21 ± 0.45 ^d^
PLA + 20% OLA + 3% HNTs	46.9 ± 1.0 ^e^	106.5 ± 0.4 ^e^	142.1 ± 0.4^e^ / 150.2 ± 0.8 ^a^	19.13 ± 0.9 ^e^	22.98 ± 0.7 ^e^	31.82 ± 0.84 ^e^
PLA + 20% OLA + 6% HNTs	46.8 ± 0.8 ^e^	103.6 ± 0.7 ^de^	141.7 ± 0.8^e^ / 149.7 ± 0.3 ^a^	20.75 ± 0.6 ^e^	22.26 ± 0.8 ^e^	31.71 ± 0.62 ^e^
PLA + 20% OLA + 9% HNTs	47.3 ± 0.9 ^e^	108.1 ± 0.8 ^e^	141.1 ± 0.7^e^ / 149.4 ± 0.6 ^a^	20.50 ± 0.5 ^e^	21.55 ± 0.5 ^e^	31.57 ± 0.73 ^e^

^a–e^ Different letters in the same column indicate a significant difference among the samples (*p* < 0.05).

**Table 4 molecules-25-01976-t004:** Main thermal degradation parameters of the polylactide (PLA)/oligomer of lactic acid (OLA)/halloysite nanotubes (HNTs) films in terms of onset temperature of degradation (*T_5%_*), degradation temperature (*T_deg_*), and residual mass at 600 °C.

Film	*T_5%_* (°C)	*T_deg_* (°C)	Residual mass (%)
PLA	342.2 ± 1.2 ^a^	383.0 ± 1.3 ^a^	1.9 ± 0.8 ^a^
PLA + 5% OLA	335.3 ± 1.0 ^b^	382.5 ± 1.0 ^a^	1.6 ± 0.6 ^b,c^
PLA + 10% OLA	313.3 ± 0.9 ^c^	382.4 ± 0.8 ^a^	1.6 ± 1.1 ^a^
PLA + 20% OLA	292.5 ± 1.2 ^d^	381.8 ± 0.9 ^a^	1.6 ± 0.9 ^c^
PLA + 20% OLA + 3% HNTs	299.5 ± 1.4 ^e^	381.0 ± 1.1 ^a^	3.1 ± 1.0 ^b^
PLA + 20% OLA + 6% HNTs	296.5 ± 0.9 ^f^	376.7 ± 0.8 ^b^	5.7 ± 0.7 ^d^
PLA + 20% OLA + 9% HNTs	282.4 ± 1.1 ^g^	348.7 ± 0.7 ^c^	8.8 ± 1.2 ^e^

^a–g^ Different letters in the same column indicate a significant difference among the samples (*p* < 0.05).

**Table 5 molecules-25-01976-t005:** Thermomechanical properties of the polylactide (PLA)/oligomer of lactic acid (OLA)/halloysite nanotubes (HNTs) films in terms of glass transition temperature (*T_g_*) and storage modulus (*E’*) measured at 30, 75, and 110 °C and

Film	*T_g_* (°C)	*E’* (MPa)		
		30 °C	75 °C	110 °C
PLA	64.1 ± 0.9 ^a^	2321.8 ± 122.1 ^a^	0.9 ± 0.1 ^a^	30.1 ± 2.8 ^a^
PLA + 5% OLA	60.7 ± 0.6 ^b^	2732.2 ± 139.2 ^b^	0.8 ± 0.2 ^a^	25.0 ± 1.6 ^b^
PLA + 10% OLA	56.0 ± 1.0 ^c^	2785.5 ± 120.2 ^b^	1.3 ± 0.2 ^b^	93.2 ± 2.9 ^c^
PLA + 20% OLA	52.0 ± 1.1 ^d^	2892.7 ± 107.4 ^c^	0.6 ± 0.1 ^a^	36.1 ± 1.9 ^d^
PLA + 20% OLA + 3% HNTs	51.2 ± 0.7 ^d^	2883.6 ± 119.3 ^c^	0.7 ± 0.1 ^a^	27.8 ± 3.0 ^b^
PLA + 20% OLA + 6% HNTs	48.4 ± 0.7 ^e^	2318.2 ± 101.5 ^a^	2.0 ± 0.2 ^c^	37.9 ± 2.7 ^e^
PLA + 20% OLA + 9% HNTs	48.3 ± 0.8 ^e^	2536.9 ± 102.1 ^f^	0.7 ± 0.1 ^a^	15.0 ± 2.1 ^f^

^a–f^ Different letters in the same column indicate a significant difference among the samples (*p* < 0.05).

**Table 6 molecules-25-01976-t006:** Water vapor permeability (WVP) and limonene permeability (LP) of the polylactide (PLA)/oligomer of lactic acid (OLA)/ halloysite nanotubes (HNTs) films.

Film	WVP × 10^14^ (kg·m/m^2^·Pa^1^·s^1^)	LP × 10^15^ (kg·m/m^2^·Pa^1^·s^1^)
PLA	1.22 ± 0.03 ^a^	3.33 ± 0.08 ^a^
PLA + 5% OLA	0.85 ± 0.02 ^b^	3.31 ± 0.07 ^a^
PLA + 10% OLA	0.65 ± 0.03 ^c^	0.94 ± 0.01 ^b^
PLA + 20% OLA	0.87 ± 0.02 ^b^	1.72 ± 0.10 ^c^
PLA + 20% OLA + 3% HNTs	0.80 ± 0.02 ^d^	0.91 ± 0.03 ^b^
PLA + 20% OLA + 6% HNTs	0.88 ± 0.03 ^b^	1.30 ± 0.09 ^d^
PLA + 20% OLA + 9% HNTs	0.88 ± 0.02 ^b^	1.42 ± 0.15 ^d^

^a–d^ Different letters in the same column indicate a significant difference among the samples (*p* < 0.05)

**Table 7 molecules-25-01976-t007:** Code and composition of each sample according to the weight content (wt%) of polylactide (PLA) and oligomer of lactic acid (OLA) in which the halloysite nanotubes (HNTs) were added as parts per hundred resin (*phr*) of blend.

Code	PLA (wt%)	OLA (wt%)	HNTs (*phr*)
PLA	100	0	0
PLA + 5% OLA	95	5	0
PLA + 10% OLA	90	10	0
PLA + 20% OLA	80	20	0
PLA + 20% OLA + 3% HNTs	80	20	3
PLA + 20% OLA + 6% HNTs	80	20	6
PLA + 20% OLA + 9% HNTs	80	20	9

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
