# Peer review of "Tailoring the Properties of Thermo-Compressed Polylactide Films for Food Packaging Applications by Individual and Combined Additions of Lactic Acid Oligomer and Halloysite Nanotubes"

_molecules, 2020, doi:10.3390/molecules25081976_

Round 1

Reviewer 1 Report

The manuscript deals with tailoring the properties of thermo-compressed polylactide films for food packaging applications by individual and combined additions of lactic acid oligomer and halloysite nanotubes.

The English language must be revised.

Abstract

This section is vague, please present your main results.

Results and discussion

Line 156- “In general, the color differences increased for the PLA/OLA films with loadings of 6 phr and 9 phr of HNTs. In particular, the ΔE*ab values were 5.15 (noticeable color difference, ΔE*ab  ≥  3.5 and < 5) and 7.68 (different colors, ΔE*ab  ≥  5), respectively.”???Table 1, please add different superscript letters for significant differences. Moreover, please revise the discussion in accordance.

Line 182- “Figure 3 gathers the field emission scanning electron microscopy (FESEM) images taken at 2000x of the cryo-fractured surfaces of the PLA films containing different OLA loadings.”???Some of the micrographs seem to show some artefacts. This part must be clarified.

Line 262- “Table 2 gathers the results of the tensile tests of the thermo-compressed films of neat PLA, PLA/OLA, and PLA/OLA/HNTs composites. In relation to the neat PLA, one can observe that the film showed mechanical properties of a hard but brittle material with values of tensile elastic modulus (Et) of 2846.3 MPa, tensile strength at yield (σy) of 54.3 MPa, and elongation at break (εb) of 3.52%. As one expected, the addition of OLA resulted in a plasticizing effect on the PLA film, yielding an increment in the flexibility and a decrease in tensile strength.”????Table 2, please add different superscript letters for significant differences. Moreover, please revise the discussion in accordance.

Line 313- “Table 3 gathers the average values of the main thermal transitions obtained from the DSC curves. The enthalpies associated with the cold crystallization and the melting processes are also reported in the table. In the 60–70 ºC range, one can observe a step change in the base lines, which corresponds to the Tg of PLA.”???Table 3, please add different superscript letters for significant differences. Moreover, please revise the discussion in accordance.

Line 380- “The thermal stability values extracted from the TGA curves are summarized in Table 4, gathering the temperature required for a loss of weight of 5% (T5%), which is representative for the onset of degradation, the maximum degradation rate temperature (Tdeg), and the amount of residual mass at 600 ºC.”???Table 4, please add different superscript letters for significant differences. Moreover, please revise the discussion in accordance.

Line 432- “Table 5 shows the E’ values of the storage modulus measured at 30, 75, and 110 ºC, being these temperatures representative of the stored elastic energy of PLA in its three states, that is, amorphous glassy, amorphous rubber, and semi-crystalline.”???Table 5, please add different superscript letters for significant differences. Moreover, please revise the discussion in accordance.

Line 469- “The water vapor permeability (WVP) and limonene permeability (LP) of the PLA films containing OLA and HNTs are shown in Table 6. Biodegradable films for packaging applications habitually show low water barrier resistance, which strongly limits their application for food packaging purposes due to physical and chemical deterioration of foodstuff.”???Table 6, please add different superscript letters for significant differences. Moreover, please revise the discussion in accordance.

Materials and methods

Line 556- “3.3.1. Color Measurements”???Illuminant and ºobserver used????

Line 571- “3.3.2. Microscopy”???Magnification levels used??

Conclusion

Please do not repeat your results and focus on your main conclusions.

References

Please format the title of each article according to the guide for authors.

Author Response

The manuscript deals with tailoring the properties of thermo-compressed polylactide films for food packaging applications by individual and combined additions of lactic acid oligomer and halloysite nanotubes.

Q1) The English language must be revised.

A1) The English expressions and grammar were revised in the whole manuscript.

Q2) Abstract

This section is vague, please present your main results.

A2) This section was rewritten to highlight the results obtained.

Results and discussion

Q3) Line 156- “In general, the color differences increased for the PLA/OLA films with loadings of 6 phr and 9 phr of HNTs. In particular, the ΔE*ab values were 5.15 (noticeable color difference, ΔE*ab  ≥  3.5 and < 5) and 7.68 (different colors, ΔE*ab  ≥  5), respectively.”???Table 1, please add different superscript letters for significant differences. Moreover, please revise the discussion in accordance.

A3) Statistical analysis of the color properties of the films was done by ANOVA with 95% confidence interval level (p≤ 0.05), please see new section 3.4. Different superscript letters were added to Table 1 to show significant differences. In the discussion the significant differences were also indicated.

Q4) Line 182- “Figure 3 gathers the field emission scanning electron microscopy (FESEM) images taken at 2000x of the cryo-fractured surfaces of the PLA films containing different OLA loadings.”???Some of the micrographs seem to show some artefacts. This part must be clarified.

A4) We have revised this figure and we did not notice any artifact.

Q5) Line 262- “Table 2 gathers the results of the tensile tests of the thermo-compressed films of neat PLA, PLA/OLA, and PLA/OLA/HNTs composites. In relation to the neat PLA, one can observe that the film showed mechanical properties of a hard but brittle material with values of tensile elastic modulus (Et) of 2846.3 MPa, tensile strength at yield (σy) of 54.3 MPa, and elongation at break (εb) of 3.52%. As one expected, the addition of OLA resulted in a plasticizing effect on the PLA film, yielding an increment in the flexibility and a decrease in tensile strength.”????Table 2, please add different superscript letters for significant differences. Moreover, please revise the discussion in accordance.

A5) The statistical analysis was also performed to the mechanical properties and Table 2 and its discussion was modified accordingly.

Q6) Line 313- “Table 3 gathers the average values of the main thermal transitions obtained from the DSC curves. The enthalpies associated with the cold crystallization and the melting processes are also reported in the table. In the 60–70 ºC range, one can observe a step change in the base lines, which corresponds to the Tg of PLA.”???Table 3, please add different superscript letters for significant differences. Moreover, please revise the discussion in accordance.

A6) The statistical analysis was also performed to the DSC results and Table 3 and its discussion was modified accordingly.

Q7) Line 380- “The thermal stability values extracted from the TGA curves are summarized in Table 4, gathering the temperature required for a loss of weight of 5% (T5%), which is representative for the onset of degradation, the maximum degradation rate temperature (Tdeg), and the amount of residual mass at 600 ºC.”???Table 4, please add different superscript letters for significant differences. Moreover, please revise the discussion in accordance.

A7) The statistical analysis was also performed to the TGA results and Table 4 and its discussion was modified accordingly.

Q8) Line 432- “Table 5 shows the E’ values of the storage modulus measured at 30, 75, and 110 ºC, being these temperatures representative of the stored elastic energy of PLA in its three states, that is, amorphous glassy, amorphous rubber, and semi-crystalline.”???Table 5, please add different superscript letters for significant differences. Moreover, please revise the discussion in accordance.

A8) The statistical analysis was also performed to the thermomechanical properties  and Table 5 and its discussion was modified accordingly.

Q9) Line 469- “The water vapor permeability (WVP) and limonene permeability (LP) of the PLA films containing OLA and HNTs are shown in Table 6. Biodegradable films for packaging applications habitually show low water barrier resistance, which strongly limits their application for food packaging purposes due to physical and chemical deterioration of foodstuff.”???Table 6, please add different superscript letters for significant differences. Moreover, please revise the discussion in accordance.

A9) The statistical analysis was also performed to the permeability results and Table 6 and its discussion was modified accordingly.

Materials and methods

Q10) Line 556- “3.3.1. Color Measurements”???Illuminant and ºobserver used????

A10) The requested information was added in Section 3.3.1.

Q11) Line 571- “3.3.2. Microscopy”???Magnification levels used??

A11) Since TEM and SEM images were taken at different magnifications, each Figure includes the magnification used in the  captions.

Q12) Conclusion

Please do not repeat your results and focus on your main conclusions.

A12) This section was rewritten to avoid repetition.

Q13) References

Please format the title of each article according to the guide for authors.

A13) The reference format was modified to match the guidelines of the journal.

Reviewer 2 Report

Dear Authors,

The manuscript deals with well written topic. Results are presented clearly and well discussed along the document. Finally, the findings support the conclusions. According to that I have to designate the paper as needed minor revision before publication in Molecules. Therefore, I recommend the acceptance of this manuscript in the current version.

Minor comments:

Table 2, 3,4,5,6: Why the authors did not do statistical analysis of the results?

Materials and Methods: the number of Eq of color measurement should be 1 and of Eq of Thermal Test should be 2 (line 563 and 595)

Author Response

The manuscript deals with well written topic. Results are presented clearly and well discussed along the document. Finally, the findings support the conclusions. According to that I have to designate the paper as needed minor revision before publication in Molecules. Therefore, I recommend the acceptance of this manuscript in the current version.

Minor comments:

Q1) Table 2, 3,4,5,6: Why the authors did not do statistical analysis of the results?

A2) As requested, statistical analysis of the color, mechanical, thermal, thermomechanical, and barrier properties of the films was done by ANOVA with 95% confidence interval level (p≤ 0.05), please see new section 3.4.

Q2) Materials and Methods: the number of Eq of color measurement should be 1 and of Eq of Thermal Test should be 2 (line 563 and 595)

A2) The equation numbers were amended.

Round 2

Reviewer 1 Report

The manuscript was improved.